


**Iron "Ore" Nothing: Benthic iron fluxes from the oxygen-deficient Santa Barbara Basin**
**enhance phytoplankton productivity in surface waters**
**De'Marcus Robinson[1*], Anh L.D. Pham[1], David J. Yousavich[2], Felix Janssen[3], Frank**
**Wenzhöfer[3], Eleanor C. Arrington[4], Kelsey M. Gosselin[5], Marco Sandoval-Belmar[1],**
**Matthew Mar[1], David L. Valentine[4], Daniele Bianchi[1], Tina Treude[1,2*]**
[1]Department of Atmospheric and Oceanic Sciences, University of California Los Angeles, Los
Angeles, CA, USA
[2]Department of Earth, Planetary, and Space Sciences, University of California Los Angeles, Los
Angeles, CA, USA
[3]HGF-MPG Joint Research Group for Deep-Sea Ecology and Technology, Alfred Wegener
Institute, Helmholtz Centre for Polar and Marine Research, Bremerhaven, Germany
[4]Department of Earth Science and Marine Science Institute, University of California, Santa
Barbara, CA 93106, USA
[5]Interepartment Graduate Program in Marine Science, University of California, Santa Barbara, CA
93106, USA
*Correspondence: De'Marcus Robinson, demarcus1.robinson@atmos.ucla.edu; Tina Treude,
ttreude@g.ucla.edu



**Abstract**
The trace metal iron (Fe) is an essential micronutrient that controls phytoplankton productivity,
which subsequently affects the cycling of macronutrients. Along the continental margin of the U.S.
West Coast, high benthic Fe release has been documented, in particular from deep anoxic basins
in the Southern California Borderland. However, the influence of this Fe release on surface
primary production remains poorly understood. In the present study from the Santa Barbara Basin,
in-situ benthic Fe fluxes were determined along a transect from shallow to deep sites in the basin.
Fluxes ranged between 0.23 and 4.9 mmol m$^{-2}$ d$^{-1}$, representing some of the highest benthic Fe
fluxes reported to date. To investigate the influence of benthic Fe release from the oxygen-deficient
deep basin on surface phytoplankton production, we combined benthic flux measurements with
numerical simulations using the Regional Ocean Model System coupled to the Biogeochemical
Elemental Cycling model (ROMS-BEC). For this purpose, we updated existing Fe flux
parameterization to include new benthic fluxes from the Santa Barbara Basin. Our simulation
suggests benthic iron fluxes support surface primary production creating positive feedback on
benthic Fe release by enhancing low oxygen conditions in bottom waters. However, the easing of
phytoplankton Fe limitation near the coast may be partially compensated by increased nitrogen
limitation further offshore, reducing the efficacy of this positive feedback.



## 1. Introduction

The California Current System (CCS), located off the coasts of Washington, Oregon, and California, is a typical Eastern Boundary Upwelling System, where seasonal upwelling supports a highly diverse and productive marine ecosystem (Chavez and Messié, 2009; Carr and Kearns, 2003). The CCS can be split into three main parts: the main equatorward California Current offshore, a subsurface poleward undercurrent fringing the continental shelf, and a recirculation pattern known as the Southern California Eddy in the Southern California Bight.

In the CCS, both upwelling and large-scale circulation provide essential nutrients to the euphotic zone, where they fuel high rates of net primary production (NPP). While seasonal upwelling dominates north of Point Conception, advection by the CCS provides a major route for nutrient supply to the Santa Barbara Channel in the Southern California Bight (Bray et al., 1999). Following phytoplankton blooms, sinking and degradation of organic matter lead to oxygen consumption and widespread oxygen loss in subsurface waters (Brander et al., 2017; Chavez and Messié, 2009). Along the southern California coast, this oxygen depletion is exacerbated by regional circulation patterns that include transport of low-oxygen waters of tropical origin along the poleward undercurrent (Evans et al., 2020; Pozo Buil and Di Lorenzo, 2017). Oxygen decline is particularly apparent in deep, isolated basins such as those found in the Southern California continental borderland, where the presence of shallow sills limits ventilation of deep waters and anoxic conditions are often encountered near the bottom (Reimers et al., 1990; Goericke et al., 2015; White et al., 2019).

In the CCS, the trace metal iron (Fe) has been identified as a limiting factor for the growth of phytoplankton (Hogle et al., 2018). Fe is an essential micronutrient that has also a considerable influence on the dynamics of phosphorus and nitrogen in the euphotic zone (Tagliabue et al., 2017). Similar to other nutrients, Fe is transported to the surface by upwelling and circulation, but the supply is generally low in an oxic environment relative to other macronutrients, reflecting rapid scavenging of the insoluble iron-oxide minerals by sinking particles that eventually accumulate in the sediment (Bruland et al., 2001, 2014; Firme et al., 2003; Till et al., 2019). While early studies suggested that Fe inputs to the CCS are dominated by rivers and aeolian deposition (Biller and Bruland, 2013; Johnson et al., 2003), more recent work highlights a combination of sources,





including benthic fluxes (Severmann et al., 2010; Noffke et al., 2012; Tagliabue et al., 2017) and
ocean currents, in redistributing Fe in coastal waters (Bray et al., 1999; Boiteau et al., 2019; García-
Reyes and Largier, 2010).
Importantly, benthic release of Fe(II), the reduced and soluble form of Fe, has been recognized as
a potential source of Fe to the surface ocean along the continental shelf and slope of the CCS,
including the deep basins of the California borderland (John et al., 2012; Severmann et al., 2010).
Under hypoxic or anoxic bottom water conditions, Fe(II) produced in the sediment during
microbial organic matter degradation coupled to Fe (III) reduction diffuses across the sediment-
water interface and accumulates in the water column (Furrer and Wehrli, 1993; Dale et al., 2015;
Severmann et al., 2010). In the CCS, this benthic Fe flux is likely to exceed atmospheric deposition
(Deutsch et al., 2021a), and may ultimately make its way to the surface by upwelling and vertical
mixing, supporting high rates of photosynthesis.
The interaction between low bottom water oxygen, Fe(II) release, and transport by the ocean
circulation are particularly important in the Santa Barbara Basin (SBB), an oxygen-deficient basin
located between the Channel Islands and mainland California in the Southern California Bight.
The SBB frequently experiences seasonal anoxia in the bottom water in fall, with irregular oxygen
flushing of dense, hypoxic water below the western sill depth (470 m) during winter and spring
(Goericke et al., 2015; Sholkovitz and Soutar, 1975; White et al., 2019). This seasonal flushing
reflects either changes in upwelling strength and frequency, or changes in stratification at the sill
depth, although the exact cause of the flushing is still unclear (Goericke et al., 2015; Sholkovitz
and Gieskes, 1971; White et al., 2019). Lack of oxygen in the deeper parts of the basin support
anaerobic microbial processes in the bottom water and sediment (White et al., 2019), including
benthic Fe reduction (Goericke et al., 2015) causing the release of Fe(II) into the water column
(Severmann et al., 2010). Ventilation events that re-oxygenate the deep basin, as well as mixing
by the vigorous submesoscale circulation (Kessouri et al., 2020) could allow upwelling of this Fe
above the sill depth and ultimately to the surface, providing a linkage between benthic processes
and upper water column biogeochemistry. Increased surface primary production supported by this
Fe source would in turn drive higher remineralization and oxygen loss in deep waters, thus
providing positive feedback to benthic Fe release. However, with a dearth of benthic Fe flux
measurements in the SBB, gaps remain in our understanding of the dynamics and impact of benthic



Fe flux, particularly with respect to its magnitude, dependence on bottom water oxygen, and ability
to reach the euphotic zone and influence primary production.
In this study, we explore the connection between benthic Fe and surface primary production in the
CCS, by investigating the influence of enhanced benthic Fe fluxes from low-oxygen waters with
a combination of field observations and model experiments. We focus on the SBB, where we
provide a new set of benthic Fe flux data determined by in-situ benthic flux chamber
measurements. We combine these new observations with existing data (Severmann et al., 2010) to
revise the representation of benthic Fe fluxes in UCLA's Regional Ocean Modeling System
coupled to the Biogeochemical Elemental Cycling (ROMS-BEC) model (Deutsch et al., 2021a).
We use the model to evaluate the effect of benthic Fe fluxes on surface nutrient consumption and
NPP, and compare their impact to that of aeolian Fe deposition in the SBB and beyond.



## 2. Materials and Methods

### 2.1 Study Site

Fieldwork in the SBB was accomplished between Oct 29 and Nov 11, 2019, during the R/V Atlantis cruise AT42-19. Sampling occurred during the anoxic, non-upwelling season along one bimodal transects with six stations total at depths between 447 and 585 m (**Fig. 1, Table 1**). The map in **Fig. 1** was created using ArcGIS Ocean Basemap. The GEBCO bathymetric data source was used to add contour lines.

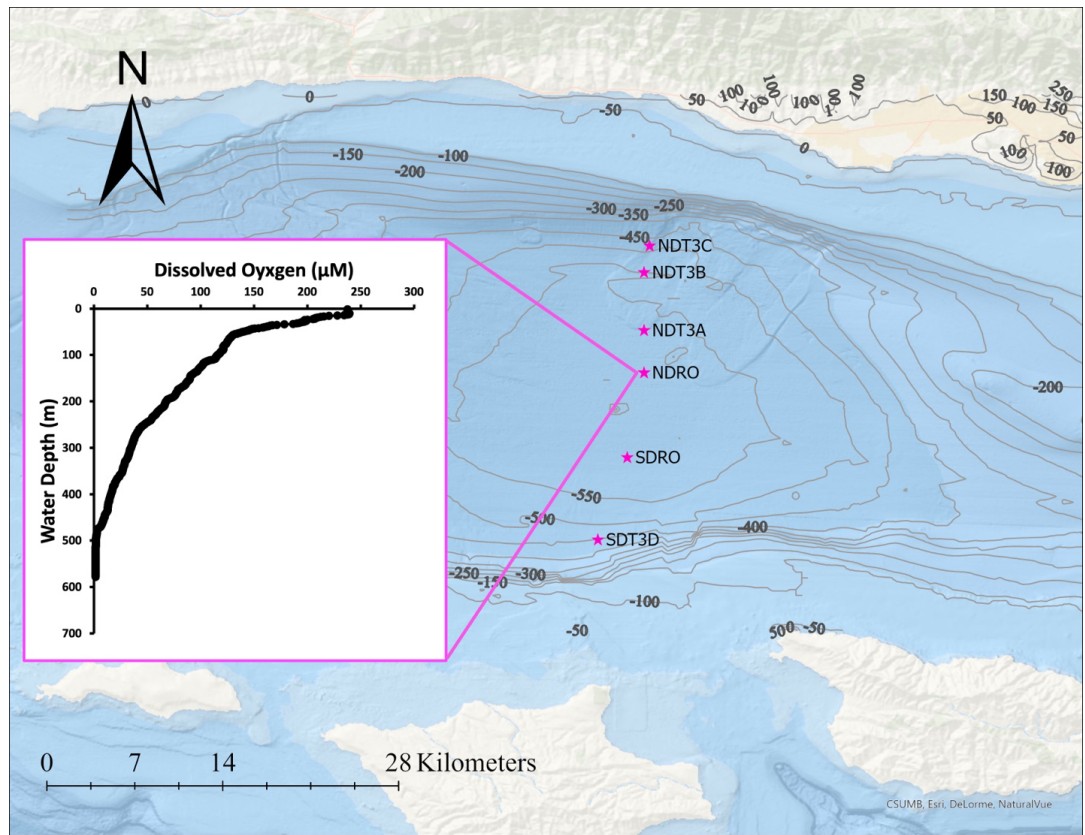

**Figure 1**. Station locations in the SBB during the AT42-19 expedition with R/V Atlantis. NDT3 (with stations A, B, C) = North Depocenter Transect Three, NDRO = North Depocenter Radial Origin, SDRO = Southern Depocenter Radial Origin, SDT3 (with station D) = Southern



Depocenter Transect Three. The small insert figure displays dissolved oxygen concentrations in
the water column at the NDRO station profiled by an optode sensor attached to the AUV Sentry.
The profile was measured at the following position: Latitude 34.2618, Longitude -120.0309.
Transects were divided into northern (NDT3 = North Depocenter Transect Three) and southern
(SDT3 = South Depocenter Transect Three) sites based on basin geography (**Fig. 1**). Stations were
labeled alphabetically from A (deepest) to D (shallowest) according to their location along the
transect, except for the deepest stations at the bottom of the basin, which were labeled Northern
Depocenter Radial Origin (NDRO) and Southern Depocenter Radial Origin (SDRO).
**2.2 Benthic Flux Chambers**
Custom-built cylindrical benthic flux chambers (Treude et al., 2009) were deployed by the ROV
Jason at the indicated stations (**Fig. 1**). Chambers were installed in a small lightweight frame made
from fiber-reinforced plastics. A stirrer was used to keep the water overlying the sediment enclosed
by the chamber well mixed. One or two replicate chambers were deployed at each site. Since
sediment in the SBB is quite soft and poorly consolidated, especially towards the deeper stations,
frames were fitted with platforms attached to the feet of the frame and with buoyant syntactic foam
to reduce sinking into the sediment. A syringe sampler equipped with 6 glass sampling syringes
withdrew 50 mL of the overlying seawater at pre-programmed times. A seventh syringe was used
to inject 50 mL of de-ionized water shortly after chambers were deployed to calculate chamber
volume from the salinity-drop recorded with a conductivity sensor (type 5860, Aanderaa Data
Instruments, Bergen, NO) in the overlying water, following the approach described in (Kononets
et al., 2021). Water samples were analyzed for Fe(II) on the ship using a Shimadzu UV-
Spectrophotometer (UV-1800), equipped with a sipper unit, following the procedure of (Grasshoff
and Ehrhardt, 1999). Fe fluxes were calculated from the slope of linear fits of Fe concentration
time series vs. time (**Fig. S1**), multiplied by the chamber volume, and divided by the surface area
of the sediment (Kononets et al., 2021).
**2.3 Numerical model (ROMS-BEC)**
To explore the impacts of benthic Fe fluxes on surface primary production, we used a well-
established ocean biogeochemical model of the CCS (Renault et al., 2016; Deutsch et al., 2021a).



The physical model component consists of the Regional Ocean Modeling System (ROMS),
(Shchepetkin, 2015; Shchepetkin and McWilliams, 2005) a primitive-equation, hydrostatic,
topography-following ocean model. As in prior work, the model domain spans the entire U.S. West
Coast, from Baja California to Vancouver Island, with a horizontal resolution of 4 km, enough to
resolve the mesoscale circulation (Capet et al., 2008). The baseline model configuration was run
over the period 1995–2017 with interannually varying atmospheric forcings. We refer the reader
to earlier publications (Renault et al., 2021; Deutsch et al., 2021a) for a complete description of
the model configuration, setup, forcings and boundary conditions used in this study.
ROMS is coupled online to the Biogeochemical Elemental Cycling (BEC) model (Moore et al.,
2004), adapted for the U.S. West Coast by (Deutsch et al., 2021b). BEC solves the equations for
the evolution of six nutrients (nitrate ($NO_3^-$), ammonium ($NH_4^+$), nitrite ($NO_2^-$), silicate ($SiO_2$),
phosphate ($PO_4^{3-}$), and iron (Fe)), three phytoplankton groups (small phytoplankton, diatoms, and
diazotrophs), a single zooplankton group, inorganic carbon, oxygen, and dissolved organic matter
(carbon, nitrogen, phosphorus, and iron). Nutrient and carbon cycles are coupled by a fixed
stoichiometry, except for silica and Fe, which use variable stoichiometries (Deutsch et al., 2021a;
Moore et al., 2001, 2004). The Fe cycle in BEC includes four separate pools: dissolved inorganic
Fe (dFe), dissolved and particulate organic Fe, and Fe associated with mineral dust. Of these, only
dissolved organic and inorganic Fe are explicitly tracked as state variables, while particulate Fe is
treated implicitly by resolving vertical sinking particle fluxes (Moore et al., 2001; Moore and
Braucher, 2008). Four main processes control the cycle of Fe: atmospheric deposition, biological
uptake and remineralization, scavenging by sinking particles, and release by sediment. The
atmospheric dFe deposition is based on the dust climatology of (Mahowald et al., 2006), and
dissolution rates from (Moore and Braucher, 2008). The benthic dFe flux is based on a compilation
of benthic flux chamber measurements on the California margin (Severmann et al., 2010) and is
parameterized as a function of the bottom water $O_2$ concentration, as discussed in (Deutsch et al.,
2021b) (see also Section 2.5). An Fe scavenging scheme removes dFe from the water column at a
rate proportional to sinking particle fluxes and dFe concentrations, assuming a uniform
concentration of 0.6 nM of Fe-binding ligands (Moore et al., 2004; Moore and Braucher, 2008).
Accordingly, scavenging rates increase rapidly at dFe concentrations greater than 0.6 nM, and
decreases rapidly below 0.5 nM (**Fig. S2**). Note that, while simplistic, this formulation is still
widely adopted by global ocean biogeochemistry models (Tagliabue et al., 2014, 2016), although





improvements have been proposed (Moore and Braucher, 2008; Aumont et al., 2015; Pham and
Ito, 2019, 2018).
As shown in previous work, the model captures the main patterns of physical and biogeochemical
variability in the CCS, providing a representation of nutrient cycles and NPP in good agreement
with observations (Renault et al., 2021; Deutsch et al., 2021b). In this paper, we further evaluate
the model solution against an extended set of dissolved Fe measurements for the CCS (see Sections
**2.4** and **3.1**).
**2.4 Fe dataset along the U.S. West Coast**
For evaluating the model ability to capture observed patterns in dFe, we compiled a dataset of the
measurements of dFe concentration along the U.S. West Coast based on published studies. These
include a global compilation (Tagliabue et al., 2016), regional programs such as CalCOFI, CCE-
LTER, IRNBRU and MBARI cruises (Bundy et al., 2016; Hogle et al., 2018; Johnson et al., 2003;
King and Barbeau, 2011), and additional independent studies (Biller and Bruland, 2013; Boiteau
et al., 2019; Bundy et al., 2014, 2015, 2016; Chappell et al., 2019; Chase, 2002; Chase et al., 2005;
Firme et al., 2003; Hawco et al., 2021; John et al., 2012; Till et al., 2019). For the final compilation,
we define dFe as the sum of the truly dissolved Fe and the dissolvable Fe, following the definitions
used in each publication. Different studies used varying filter sizes to define the dFe pool, with
0.20, 0.40, and 0.45 µm as the most common. Measurement methods also vary slightly between
studies, with samples taken with bottles, pump systems and/or by surface tows. In some cases,
samples were acidified for short periods of time before analysis. Despite variable approaches, we
find a good agreement between different sets of observations and consider the merged dataset as
representative of the dFe distribution in the CCS. The final compilation includes observations from
1980 to 2021, with most of the data from the period 1997-2015, and from the upper 100 m of the
water column.
**2.5 Experimental Design**
To evaluate the impact of Fe fluxes from low-oxygen sediment in the SBB on surface
biogeochemistry, we designed a suite of model sensitivity experiments with ROMS-BEC in which



external sources of Fe are modified relative to a baseline simulation. Accordingly, we run the
following model experiments:
**_Control_**: This is the baseline model simulation, using the Fe flux parameterization based on the
measurements by (Severmann et al., 2010) following the parameterization by (Deutsch et al.,
2021b). Fe release follows the equation:
$\log_{10}\Phi(Fe) = 2.5 - 0.0165 \cdot O_2$ (**Eq. 1**)
where $O_2$ is the concentration of oxygen in mmol m$^{-3}$ and $\Phi(Fe)$ is the Fe flux in µmol m$^{-2}$ d$^{-1}$.
Note that this experiment reflects the original Fe flux parameterization in UCLA's ROMS-BEC
and does not include information from the Fe flux measurements conducted during AT42-19,
which show significantly higher Fe release under anoxic conditions.
**_Low Oxygen Threshold_**: The purpose of this experiment is to evaluate the importance of enhanced
Fe fluxes under low-oxygen conditions in the bottom water. Benthic Fe fluxes are calculated as in
**_Control_**, but capped at a constant value when oxygen decreased below a specific threshold. We
performed two "**_Low Oxygen Threshold_**" model experiments. The first uses an $O_2$ threshold of 100
µM (**_Low Oxygen Threshold-100_**), and caps Fe release at 0.85 µmol m$^{-2}$ d$^{-1}$ when oxygen drops
below 100 µM. The second uses a threshold of 65 µM (**_Low Oxygen Threshold-65_**), and caps Fe
release at 1.48 µmol m$^{-2}$ d$^{-1}$ when oxygen drops below 100 µM. The 65 µM threshold is close to
the typical definition of hypoxia (~60 µM), while the 100 µM threshold was chosen to investigate
the general impact of benthic Fe fluxes from low-$O_2$ coastal sediment, because around 80 % of the
shelf in our model is characterized by bottom $O_2$ concentration lower than 100 µM (**Fig. S3**)
**_High-flux_**: This simulation investigates the importance of high benthic Fe fluxes in the SBB, and
is based on the new benthic measurements from AT42-19 combined with previous observations
(Severmann et al., 2010). We derived and applied a new parameterization for the dependence of
benthic Fe flux on bottom $O_2$ using the combined Fe flux dataset:
$\log_{10}\Phi(Fe) = 2.86 - 0.01 \cdot O_2$ (**Eq. 2**)
This revised formulation is only applied in the SBB, while the same formulation as **_Control_** is used
elsewhere. We further corrected a model bias that limits simulations to $O_2$ concentrations >30



mmol m$^3$. This correction is crucial to allow the model the estimation of benthic Fe fluxes under
anoxic conditions, rather than simulating fluxes at 30 mmol m$^3$. We therefore applied a constant
deduction of 30 mmol $O_2$ m$^{-3}$ to the model's bottom water $O_2$ based on the average difference
between model and observed $O_2$ in the SBB.
***Dust-off***: The purpose of this experiment is to evaluate the importance of aeolian Fe deposition in
the CCS. In this experiment, the atmospheric Fe deposition is set to zero; all other settings are
identical to the *Control* experiment.
Each model sensitivity experiment is run separately over a time frame of 6 years from 2004-2009,
using the same set of forcings and initial conditions. Results from the final year (2009) are analyzed
by comparing differences in biogeochemical fields (Fe, $NO_3^-$, and NPP) to results from the *Control*
run.



## 3. Results

### 3.1 In-situ benthic Fe fluxes and model parameterization

Benthic Fe fluxes from in-situ benthic chamber measurements during the AT42-19 expedition are shown in **Table 1**. High Fe flux was recorded at the anoxic depocenter stations (4.90 and 3.92 mmol $m^{-2}$ $d^{-1}$ at SDRO and 3.49 mmol $m^{-2}$ $d^{-1}$ at NDRO). Fe fluxes at the shallower hypoxic stations (NDT3C, NDT3B, and SDT3D) were an order of magnitude lower in comparison. The Fe flux at the hypoxic NDT3A station was approximately half the flux of the depocenter.

**Table 1.** Station details and geochemical parameters determined during the AT42-19 expedition. Benthic Fe fluxes were determined using in-situ benthic chambers. Dissolved oxygen concentrations were measured in the water column at 10 m above the seafloor using a Seabird optode sensors attached to the ROV Jason. At stations with two benthic chamber deployments (NDT3A and SDRO), $O_2$, coordinates, and depth were averaged as there were only minimal differences between the two chamber deployments.

| Station | Fe Flux [mmol $m^{-2}$ $d^{-1}$] | Oxygen [μM] | Latitude | Longitude | Depth [m] |
|---|---|---|---|---|---|
| NDT3C | 0.23 (n=1) | 5.3 | 34.3526 | -120.0160 | 499 |
| NDT3B | 0.36 (n=1) | 6.8 | 34.3336 | -120.0188 | 535 |
| NDT3A | 1.73; 1.20 (n=2) | 9.6 | 34.2921 | -120.0258 | 572 |
| NDRO | 3.49 (n=1) | 0.0 | 34.2618 | -120.0309 | 581 |
| SDRO | 4.90; 3.92 (n=2) | 0.0 | 34.2011 | -120.0446 | 586 |
| SDT3D | 0.58 (n=1) | 9.6 | 34.1422 | -120.0515 | 446 |

Trends in the Fe fluxes suggest modulation by oxygen concentration, water depth, and/or bathymetry. We observed a decrease in the Fe flux with a decrease in water depth (**Fig. 2**). There was also a slight trend of higher Fe fluxes with lower $O_2$ concentrations (most pronounced when oxygen reaches zero); however, since oxygen concentrations were relatively low at all stations (<10 μM) it is difficult to distill a clear pattern based on the small dataset. Notably, the NDT3A station showed a high Fe flux despite exhibiting the same oxygen concentration as the shallower station SDT3D. Basin bathymetry may also contribute to observed differences in the flux. For instance, the deeper depocenter and A-station showed higher averaged fluxes than the B, C, and D



stations. We further noticed differences between the north and south extension of the transect. The
southern stations (SDRO and SDT3D) showed a higher Fe flux than the northern stations (NDRO
and NDT3C).

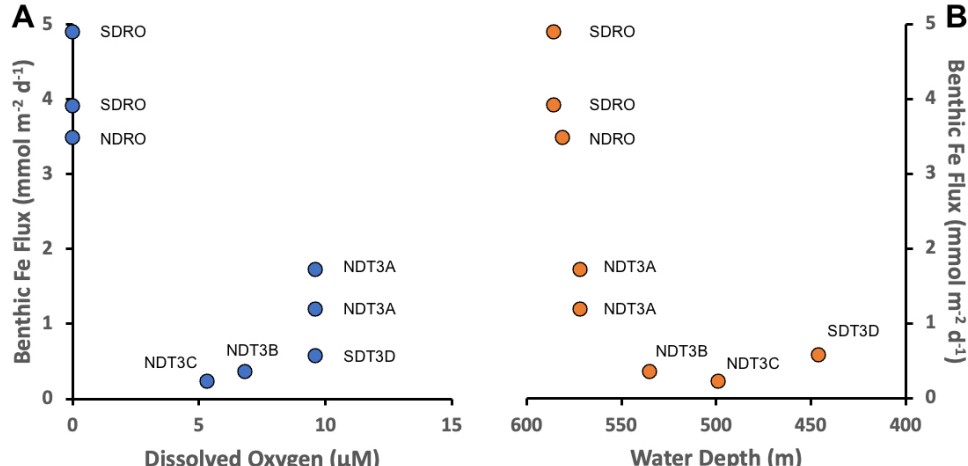


**Figure 2**. Benthic in-situ Fe fluxes. A: Fluxes as a function of oxygen. B: Fluxes as a function of
(station) water depth. Note that water depth is shown in reverse order. For station details see Table

276    1.

We combined Fe fluxes determined during AT42-19 with previous determinations along the CCS,
as compiled by (Severmann et al., 2010), and analyzed them as a function of bottom water oxygen
(**Fig. 3**). Pooled together, the measurements can be well described by an exponential increase of
Fe fluxes with declining bottom water oxygen (Severmann et al., 2010), consistent with the Fe
flux parameterization adopted in the ROMS-BEC model (Deutsch et al., 2021b). Several
observations from the AT42-19 cruise (red dots in **Fig. 3**) exceed the range of previous
measurements (yellow dots in **Fig. 3**), likely owing to the anoxic or near-anoxic conditions in the
water. Relative to the exponential fit to the dataset by (Severmann et al., 2010) (green line in **Fig.**
**3**) the revised fit to the pooled data (purple line in **Fig. 3**) expands Fe fluxes by approximately two
times at oxygen concentrations near zero, and up to one order of magnitude at concentrations near
100 µM.



## 3.2 Model evaluation

The *Control* simulation captures the magnitude and patterns of the observed dFe distribution in the upper ocean (**Fig. 4**), consistently with our knowledge of the ocean Fe cycle. In both model and observations, dFe concentrations are low at the surface, because of phytoplankton uptake, and increase gradually in subsurface waters due to remineralization and benthic fluxes (**Fig. S4**). The highest dFe concentrations are found along the coast, likely related to high surface productivity and carbon export combined with basin bathymetry and oxygen deficiency. In the open ocean, dFe concentrations are low in both model and observations, reflecting a combination of phytoplankton uptake, scavenging, and low external inputs.

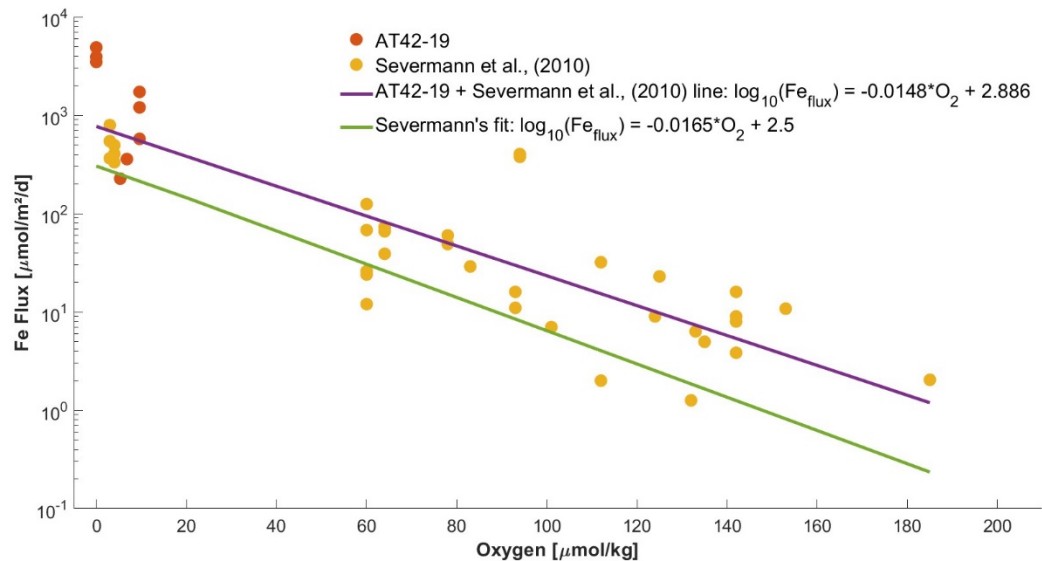

**Figure 3**. Combined Fe flux data as a function of oxygen. Fe flux data from (Severmann et al., 2010) (orange dots) were fitted with a line of best fit (green) as the original model parametrization. AT42-19 (red dots) were fitted along a line of best fit (purple) that includes Severmann's data point.

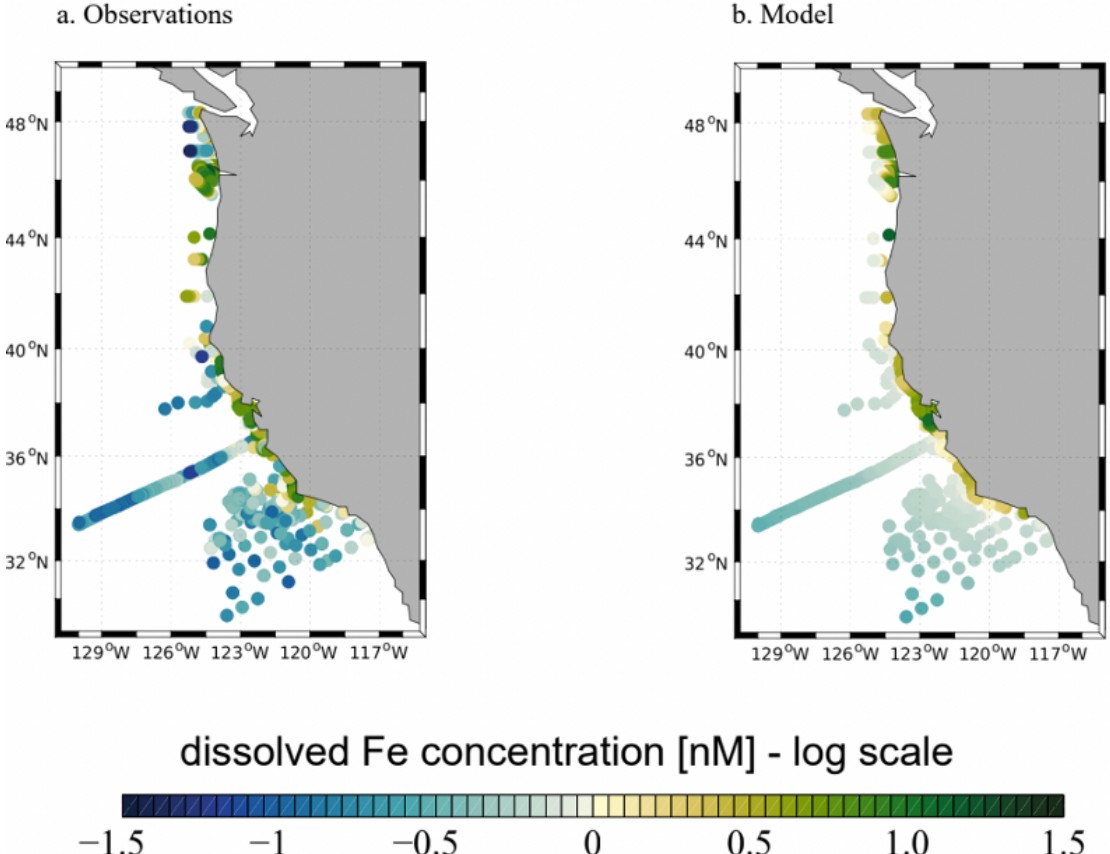

302

**Figure 4**. (a) Observed dFe concentrations (nM) from the U.S. West Coast compilation (see section 2.4) averaged between 0 and 100 m depth. (b) Modeled annual dFe concentrations (nM) averaged between 0 and 100 m depth (note that locations are identical to (a)).

The agreement of the model dFe with observations (R=0.5, **Fig. 4b**) reflects results from other ocean models compiled in (Tagliabue et al., 2016). However, the model tends to underestimate the sharp dFe gradient between coastal and open ocean waters, overestimating dFe in the open ocean and producing too uniform concentrations offshore and at depth (**Fig. 4; Fig. S4**). These biases are likely related to the simple Fe scavenging scheme, which assumes a constant Fe-binding ligand concentration of 0.6 nM. The low number and episodic nature of in-situ measurements may also explain some of the mismatch between model and observations.



At the scale of the CCS, the *Control* simulation produces lower surface dFe in the southern domain
(33 - 36°N), and higher surface concentration in the northern domain (40 - 45°N) and near the
central coast (**Fig. 5a**). While these patterns reflect a combination of internal Fe cycling and
external inputs, the elevated dFe in the northern part of the CCS, in particular offshore, can be
partly attributed to higher aeolian deposition in that region (**Fig. S5**) as well as coastal inputs from
the Juan De Fuca strait (Deutsch et al., 2021b).
Relative to Fe, $NO_3^-$ shows fewer variable patterns along the meridional direction, and a more
pronounced signature of coastal upwelling, with higher concentrations nearshore in the central
coast (36 – 40°N), and low concentrations in the Southern California Bight and in offshore waters
(**Fig. 5b**). The signature of upwelling is also apparent in NPP (**Fig. 5c**), with high values near the
coast, in particular in the central region, and decreasing values offshore. These patterns are
consistent with observations, as discussed in prior work (Deutsch et al., 2021b)

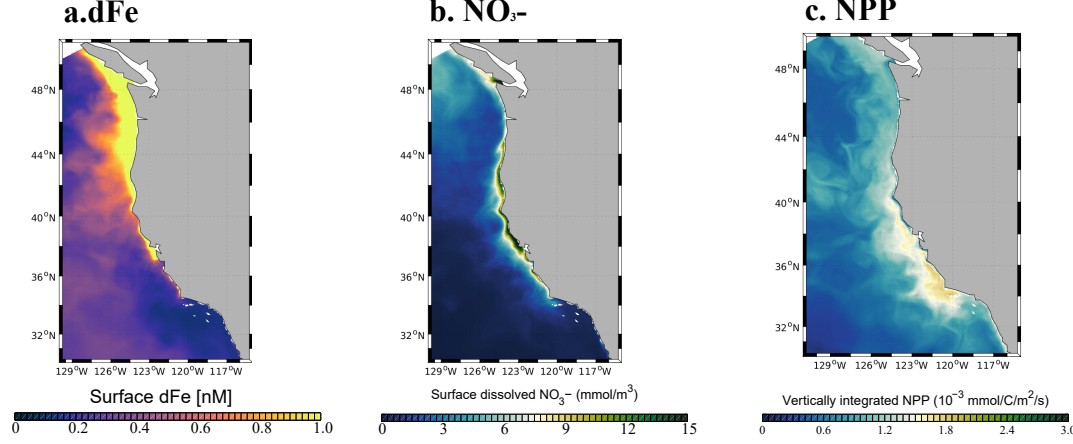



**Fig. 5.** Surface dFe concentration (a), surface $NO_3^-$ concentration (b), and vertically integrated net
primary production (NPP) (c) from the *Control* model run.
**3.3 Low Oxygen Threshold: Impact of benthic Fe flux from low-oxygen bottom water**
We quantify the importance of benthic Fe fluxes from low-oxygen bottom water by analyzing
results from the *Low Oxygen Threshold* experiments, in which we cap the high benthic Fe flux
when oxygen declines below a given threshold (**Fig 6**).



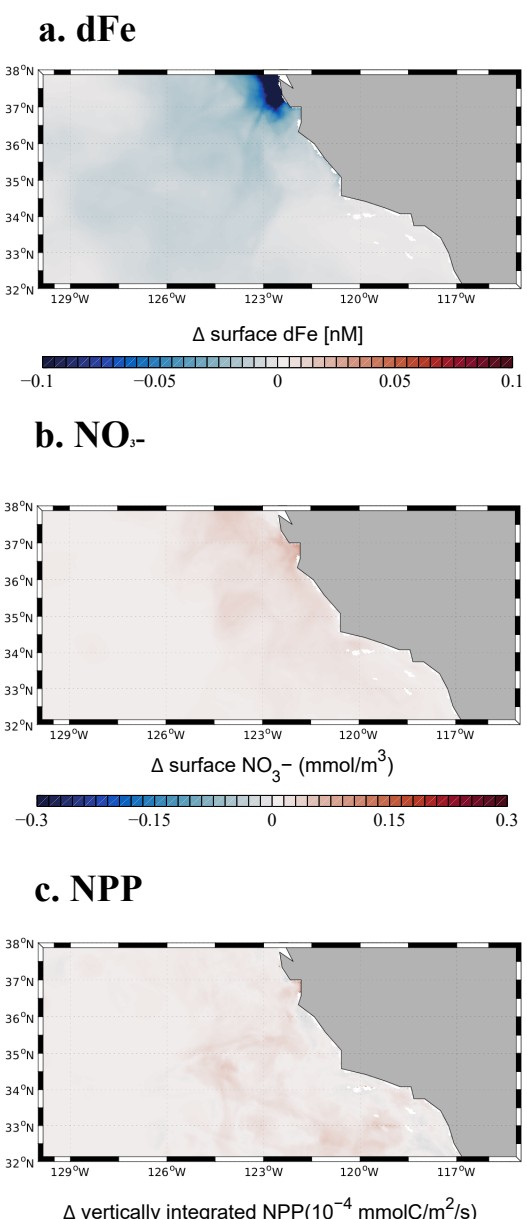



**Figure 6**. (a) Surface dFe anomalies, (b) Surface $NO_3^=$ anomalies, and (c) vertically integrated net
primary production (NPP) from the *Low Oxygen Threshold-100* model run relative to the *Control*
model run. Graphs focus on areas around the SBB.



As expected, the forced decrease in benthic Fe flux in the *Low Oxygen Threshold-100* simulation
leads to a significant decrease in the surface dFe concentration (**Fig. 6a**). This decrease is
particularly strong along the coast, but also extends into the open ocean (**Fig S6**). This trend
indicates that dFe released from low-oxygen sediment is effectively transported to the surface and
offshore, where it can affect primary production. Consistently, the decrease in surface dFe drives
a decline in NPP near the coast (**Fig. 6c**), where phytoplankton rely the most on benthic-derived
Fe. However, NPP also shows a significant increase offshore, especially between 32N and 36N.
This increase can be explained by the relative importance of Fe vs. N limitation along a cross-
shore productivity gradient. While near the coast phytoplankton are frequently Fe limited (up to
50% of the time in the model), they tend to be almost exclusively N-limited moving offshore
(Deutsch et al., 2021b). As Fe limitation reduces NPP near the coast in the *Low Oxygen Threshold-*
*100* experiment, $NO_3^-$ utilization also declines, so that more $NO_3^-$ can accumulate in surface waters
(**Fig. 6b**). Shallow transport of excess $NO_3^-$ by Ekman transport and eddies can fertilize offshore
waters, releasing N limitation and fueling an increase in NPP away from the coast (**Fig. 6c**).
In the *Low Oxygen Threshold-65* simulation, the patterns of response are similar to the *Low Oxygen*
*Threshold-100* simulation. However, the magnitude of the response is smaller, as only about 50 %
of the CCS coast is characterized by bottom $O_2$ concentrations below 65 µM, as compared to 80
% for $O_2$ below 100 µM. Hence, the decrease in benthic Fe release, and its cascading effects on
surface Fe, $NO_3^-$ and NPP are more muted in this simulation (**Fig. S7**).
**3.4 High-Flux: Impact of high Fe flux from updated parametrization on NPP, surface dFe,**
**and surface $NO_3^-$**
The *High-flux* experiment quantifies the impact of the higher benthic Fe fluxes determined during
the AT42-19 cruise in the SBB. In this experiment, we observe a dramatic increase of surface dFe
along the coast, both within the Santa Barbara Channel, and north of it (**Fig. 7, Fig. S8**). This
increase leads to a slight depletion of $NO_3^-$ along the coast (by less than 1 µM), and a patchwork
of changes in NPP, with a general increase nearshore and in the southern section of the Southern
California Bight, and a decrease offshore. These patterns are opposite in sign to the changes
observed in the *Low Oxygen Threshold* experiments, although more intense, and can be explained
by similar dynamics. Nearshore, where Fe is more frequently limiting, higher Fe availability





releases Fe limitation and drives the higher NPP and more intense $NO_3^-$ drawdown. Further
offshore, where waters tend to be more N limited, a reduced supply of $NO_3^-$ decreases NPP.
Interestingly, the localized increase in Fe fluxes from the deep SBB has cascading effects on NPP
across a much larger region in the CCS. This indicates that Fe released at depth from the anoxic
basins is upwelled or mixed to the ocean surface and re-distributed by the oceanic circulation, both
northward along the coast, and southward into the Southern California Bight, likely by
recirculation within the Southern California Eddy.
**3.5 Dust-off: Role of atmospheric Fe deposition**
We evaluate the importance of aeolian Fe sources in the *Dust-off* simulation, in which atmospheric
Fe deposition is set to zero. In this experiment, surface dFe decreases everywhere in the CCS, but
the decrease is particularly evident in the open ocean and the northern part of the domain (**Fig. 8a**).
This Fe decrease leads to a widespread decrease in NPP in the northern part of the domain (40N
to 48N, see **Fig. S9**), with stronger negative anomalies away from the coast. The decline in NPP
is accompanied by a broad decrease in $NO_3^-$ utilization, particularly evident offshore, where
phytoplankton rely mostly on Fe delivery by dust. In contrast, we observe a broad increase in NPP
in the southern part of the domain (south of 40°S) and in coastal areas, likely reflecting increased
availability of $NO_3^-$ transported southward by the broad California Current. However, the
relatively weak magnitude of NPP responses to changes in dust deposition demonstrate that
phytoplankton in the coastal areas and the southern CCS rely more on Fe delivery by benthic
sources as compared to atmospheric deposition (**Table S1**).



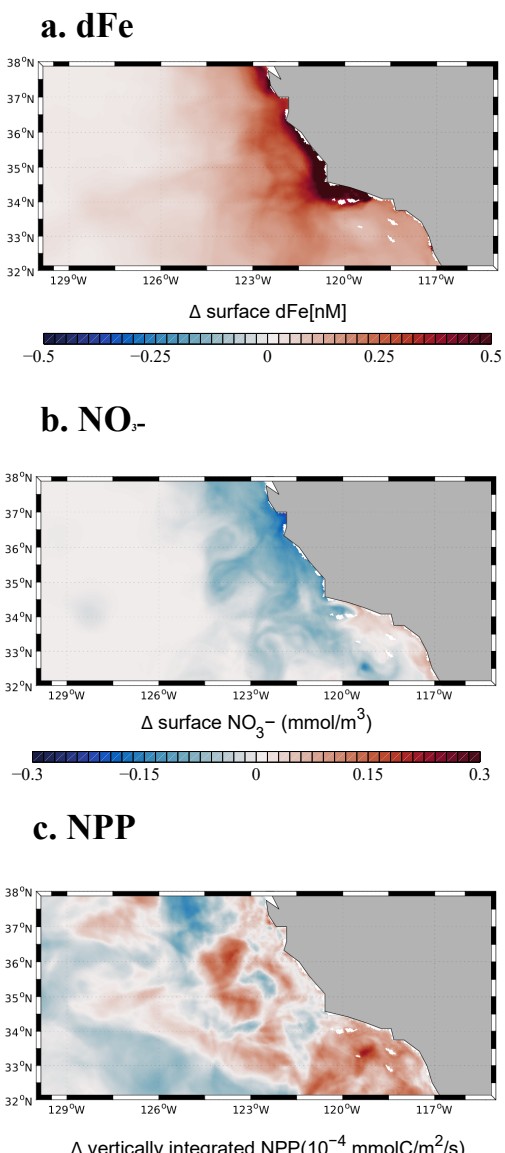


**Figure 7.** Surface dFe anomalies (a), Surface NO₃⁻ anomalies (b), and vertically integrated net
primary production (NPP) anomalies (c) from the *High-flux* model run relative to the *Control*
model run. The graphs focus on areas around the SBB. For the full model domain of the U.S West
coast see **Fig. S8**.




## a. dFe

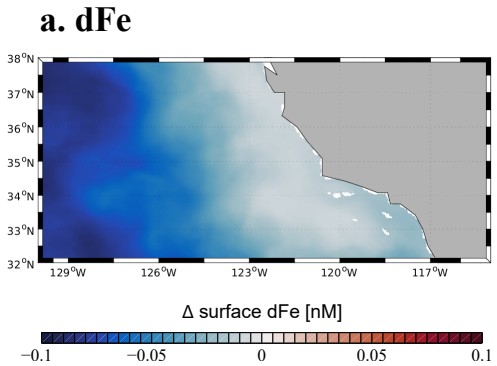

## b. NO₃-

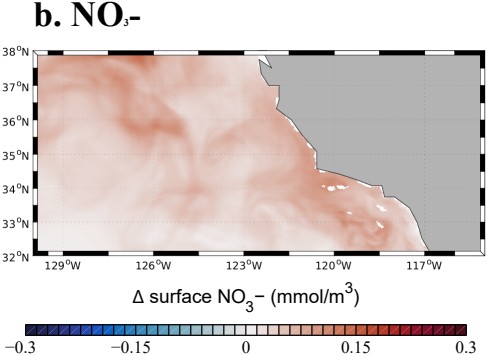

## c. NPP

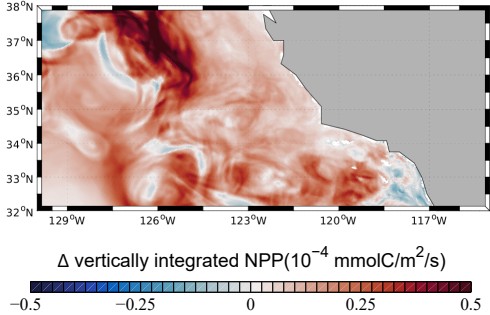


**Figure 8.** Surface dissolved Fe (a), surface NO₃⁻ anomalies (b), and vertically integrated net
primary production (NPP) (c) from the *Dust-off* model run relative to the *Control* model run. The
graphs focus on the areas around the SBB. For the full model domain of the U.S West coast see
**Fig. S9**.





## 4. Discussion

### 4.1 Benthic Fe flux feedbacks on SBB biogeochemistry

The influence of bottom water oxygen concentration on the exchange of solutes between the sediment and the water column has been well documented (Soetaert et al., 2000; Sommer et al., 2016; Testa et al., 2013). Under hypoxic or anoxic bottom water conditions, organic matter sedimentation sustains anaerobic respiration at the sediment-water interface and in the sediment (Furrer and Wehrli, 1993; Middelburg and Levin, 2009). Reduced compounds accumulate in pore waters forming chemical gradients (Widdows and Brinsley, 2002) that result in the flux of solutes such as Fe(II) out of the sediment, and their accumulation in bottom water (Jørgensen and Nelson, 2004; McMahon and Chapelle, 1991; Middelburg and Levin, 2009; Yao et al., 2016). Similar conditions are observed in the SBB, where high sedimentation rates, water column denitrification below the sill depth, and high pore-water concentrations of sulfide and Fe(II) have been observed (Behl and Kennett, 1996; Bray et al., 1999; Goericke et al., 2015; Sholkovitz and Soutar, 1975; Sigman et al., 2003; White et al., 2019).

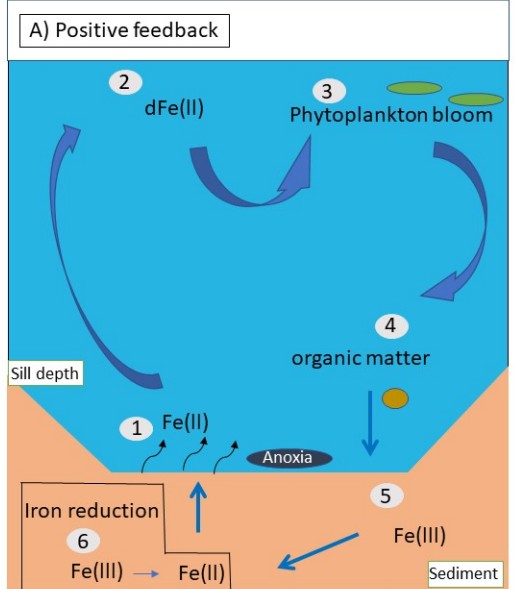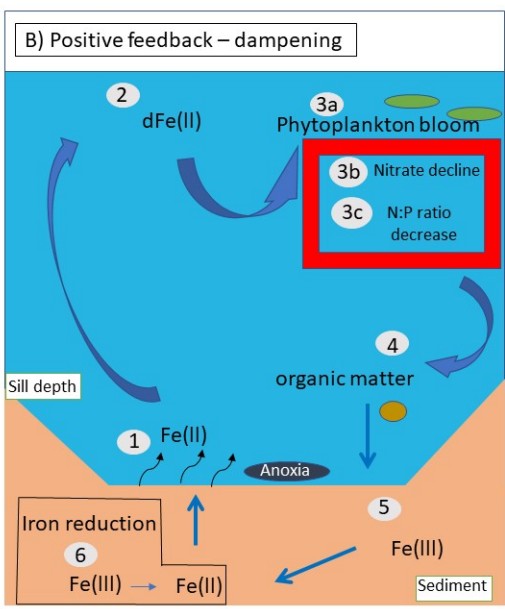

**Figure 9.** Positive feedback loop in the SBB (A): 1. Benthic Fe release into the anoxic (<3 µM), or severely hypoxic (3-20 µM) bottom water. 2. Upwelled Fe reaches the surface ocean





contributing to dFe. 3. Dissolved Fe is assimilated by phytoplankton producing phytoplankton
blooms, organic matter and siderophores at the surface. 4. Organic matter is exported from the
surface to the deep ocean. 5. Organic matter accumulates at the sediment-water interface. 6. During
remineralization of organic matter, iron-reducing bacteria reduce Fe(III) to Fe(II). Negative
feedback loop in the SBB (B): 1-3 (not including 3 b and c) and 4-6 are identical to (A). Part 3b
and 3c shows the decline of $NO_3^-$ from the amplification of dFe, which causes a decrease in the
N:P ratio.
The intense flux of dFe from the sediment suggests the potential for positive biogeochemical
feedbacks in the SBB and more broadly in the CCS (**Figs. 6 – 8**). However, our simulations also
indicate the presence of complex biogeochemical responses between Fe, $NO_3^-$ and NPP that may
dampen the effects of these feedbacks.
Under a positive feedback scenario (**Fig. 9a**), anoxic and nearly anoxic bottom water conditions
facilitate Fe(II) diffusion from the sediment into the bottom water. In the SBB, this Fe eventually
reaches the surface via upwelling and mixing processes, which are likely enhanced in the presence
of complex bathymetry and islands (Kessouri et al., 2020). This additional dFe input fertilizes
coastal waters and increases primary production. Newly formed organic matter eventually sinks
towards the seafloor as a rain of organic particles, supporting low-oxygen concentrations in the
bottom water, and fueling anaerobic respiration, including Fe reduction, in the sediment. This
chain of processes thus represents a positive feedback loop that maintains high Fe(II) release from
the sediment, as long as the bottom water remains hypoxic or anoxic (Mills et al., 2004; Noffke et
al., 2012; Sañudo-Wilhelmy et al., 2001; Dale et al., 2015). However, our simulations suggest that
this positive feedback loop is dampened by increased $NO_3^-$ limitation under higher Fe supply (**Fig.**
**9b**), which would limit the increase in NPP. Transport of N-depleted coastal waters reduces NPP
offshore (**Fig. 7**), further counteracting the positive feedback loop.
Additional processes may dampen or alter this feedback loop. Increased anoxia in bottom water
and sediment favors the removal of fixed N by denitrification (Goericke et al., 2015; White et al.,
2019). Upwelling of $NO_3^-$-depleted waters would then reduce surface productivity by increasing
N limitation (Gruber and Deutsch, 2014). Release of Fe(II) from the sediment could also impact
phosphate dynamics in the SBB. Phosphate is scavenged by iron during oxidation of Fe(II) in the





water column and sediment because of the ability of Fe(III) minerals to bind phosphate. After
burial, phosphate is released due to reduction of solid Fe(III) minerals to dissolved Fe(II), and
diffuses upward to be either re-adsorbed by Fe(III) at the oxic sediment-water interface, or released
to the bottom water under anoxic conditions (Dijkstra et al., 2014). The latter scenario is consistent
with our in-situ benthic flux chamber measurements revealing increased phosphate releases from
the sediment with increased SBB depth (data not shown). Increased release of phosphate into the
water column, and transport to the surface, could decreases the N:P ratio of phytoplankton,
especially downstream of waters where denitrification occurred (Deutsch et al., 2007). In the
presence of N limitation, these conditions could favor the activity of nitrogen-fixing
microorganisms (Mills et al., 2004; Noffke et al., 2012; Sañudo-Wilhelmy et al., 2001), further
modulating surface NPP (Deutsch et al., 2007).
**4.2 Contribution of physical transport on surface Fe**
Our numerical experiments suggest that Fe released into the deep SBB can reach and fertilize
surface waters. This finding highlights the critical role of bottom water upwelling and mixing in
the SBB. There is ample literature describing seasonal surface circulation and bottom water
renewal and its effect on nutrients in the SBB (Bray et al., 1999; Hendershott and Winant, 1996;
Sholkovitz and Gieskes, 1971). However, the frequency and rate of seasonal bottom water flushing
events, and the processes responsible for vertical mixing and upwelling across hundreds of meters
remain poorly understood (Shiller et al., 1985; Sholkovitz and Gieskes, 1971; White et al., 2019).
It is likely that interaction between wind-driven upwelling events and submesoscale eddies, which
are particularly intense inside the Santa Barbara Channel (Kessouri et al., 2020), favors upward
mixing of deep bottom water following flushing events.
**4.3 Quantifying expansion of anoxia in the SBB**
Changes in source waters and global oxygen loss in the Southern California Bight have contributed
to decreasing $O_2$ levels throughout the Southern California Bight and the SBB (Zhou et al., 2022).
With the outlook of a continuing decline in oceanic oxygen (Bopp et al., 2013; Kwiatkowski et al.,
2020), quantifying the expansion of hypoxic and anoxic zones in the SBB is vital to understand
the dynamics and fate of Fe(II) and other reduced compounds (e.g., ammonium ($NH_4^+$), hydrogen
sulfide ($H_2S$)) in deep low-oxygen waters. In the SBB, bottom water renewal events have





experienced a decline in frequency and magnitude, driving an expansion of hypoxic and anoxic
conditions in deep waters (White et al., 2019). This expansion leads to an increase in anaerobic
reactions, such as denitrification in the water column (White et al., 2019) as well as Fe reduction,
sulfate reduction, and dissimilatory nitrate reduction to ammonium (DNRA) in the sediment
(Valentine et al., 2016; Treude et al., 2021; Sommer et al., 2016). Expansion of low oxygen waters
could intensify the positive feedback loop between Fe release, NPP and $O_2$ loss (**Fig. 9**). However,
to date, despite the evidence for more frequent anoxia, there is no clear quantitative record of the
vertical or horizontal expansions of oxygen-deficient waters in the SBB.



**5. Conclusion**
Our field campaign in the SBB measured a remarkably high flux of Fe(II) from the sediment (0.23
– 4.9 mmol m$^{-2}$ d$^{-1}$), greater than in previous studies from this region (Severmann et al., 2010) and
from other oxygen minimum zones (Dale et al. 2015; Homoky et al. 2021). Using a series of
simulations with an ocean biogeochemical model, we show that this high Fe release from deep,
low-oxygen sediment has a significant impact on surface nutrients and productivity in the SBB
and the Southern California Bight, where Fe is often limiting (Hogle et al., 2018). We also
highlight the impacts of coastal Fe inputs on waters further offshore. While phytoplankton in
coastal areas directly benefit from Fe fertilization, increased NO$_3^-$ utilization in coastal waters can
cause N-limitation of phytoplankton further downstream in open-ocean areas. Thus, benthic Fe
fluxes can modulate Fe and NO$_3^-$ limitation in ways that partially counteract one another along the
cross-shore productivity gradient of the CCS. Our model simulations also suggest that Fe inputs
from atmospheric deposition are mostly important in the open ocean north of 40°N, where
phytoplankton rely on Fe delivery by dust. However, we also show that changes in atmospheric
Fe deposition can alter ocean productivity in the southern CCS by altering NO$_3^-$ utilization further
downstream. Our results support the idea that benthic Fe fluxes are the major source of Fe in the
southern CCS and are supplemented by atmospheric deposition in the northwestern region, leading
to relatively high NPP coastwide.
Over the entire U.S. West Coast, changes in the dependence of benthic Fe release on bottom O$_2$
can halve (*Low Oxygen Threshold-100*) or double (*High-flux*) the mean benthic Fe flux. While our
observations are based on snapshots of O$_2$ and Fe flux, they have implications for the temporal
variability of Fe supply. High benthic Fe fluxes are observed during the anoxic fall season, while
seasonal flushing in winter and spring likely decrease the flux of Fe by increasing bottom water
O$_2$ and Fe oxidation and retention near the sediment.
We suggest that benthic Fe fluxes from deep anoxic basins reach the surface ocean, contributing
to feedbacks between Fe and NO$_3^-$ limitation and NPP. Specifically, high Fe fluxes from low-
oxygen sediment support higher NPP near the coast, in turn leading to increased respiration and
O$_2$ loss at depth, maintaining high Fe release. This positive feedback loop is dampened by
increased NO$_3^-$ limitation, which reduces NPP downstream of coastal regions. This benthic-pelagic



coupling demonstrates the importance of sediment-derived Fe fluxes on the coastal ecosystem of
the CCS, and the role of vertical transport processes in connecting deep environments to surface
waters along continental margins.
We highlight the need for further studies focusing on feedbacks between benthic processes and
surface biogeochemistry. For example, fixed N loss by denitrification and enhanced release of
phosphorous under low-oxygen bottom water are likely to further modulate these interactions.
Seasonal studies based on stable isotope, radiotracer, and geochemical techniques are required to
track the fate and transport of nutrients in the SBB and similar low-$O_2$ coastal regions, shedding
light on the microbial metabolisms that influences these dynamics. Ocean biogeochemical models
for regional and global studies should incorporate new observations of benthic fluxes and their
sensitivity to bottom $O_2$ and other variables. This model adaptation would shed light on the impact
of $O_2$ variability, from seasonal to interannual and longer timescales, including the effects of long-
term oceanic $O_2$ loss, on the feedbacks between benthic nutrient fluxes and surface
biogeochemistry.



**Acknowledgements**

We thank the captain and crew of R/V Atlantis, the crew of ROV Jason, the crew of AUV Sentry, and the science party of research cruise AT42-19 for their technical and logistical support. We thank Q. Qin, M. O'Beirne, A. Mazariegos, X. Moreno, and A. Eastman for assisting with shipboard analyses. Funding for this work was provided by the US National Science Foundation, NSF OCE-1829981 (to TT), OCE-1756947 and OCE-1830033 (to DLV), and OCE-2023493 (to DB and ALP). Computational resources were provided by the Expanse system at the San Diego Supercomputer Center through allocation TG-OCE170017 from the Extreme Science and Engineering Discovery Environment (XSEDE), which was supported by National Science Foundation grant 1548562.

**Code availability**

The physical and biogeochemical codes used for our simulations can be accessed at: https://github.com/UCLA-ROMS/Code.

The model output can be accessed through Zenodo: (link will be provided before publication)

**Data availability**

In-situ benthic Fe flux data are accessible through the Biological & Chemical Oceanography Data Management Office (BCO-DMO) under the following DOI: (link will be provided before publication).

**Author contributions.**

DR, TT, DB, and AP conceived this study. DM, DJY, FJ, FW, ECA, KMG, DLV and TT conducted the sampling at sea. DJY transformed and interpreted ROV Jason data. FJ and FW constructed and managed benthic flux chambers. DYJ and DR analyzed Fe(II) and assisted with the flux calculation. MM provided the compiled Fe measurements along the U.S. West Coast. AP and MS performed the model simulations. DR, DB, AP and TT wrote the manuscript with input from all co-authors.

**Competing interests**



The authors declare that they have no known competing financial interests or personal
relationships that could have appeared to influence the work reported in this paper





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
