# Peer review of "Iron "Ore" Nothing: Benthic iron fluxes from the oxygen-deficient Santa Barbara Basin"

_Biogeosciences, 2022_

## Referee Comment (RC2)

Review Comment: bg-2022-237

The manuscript by Robinson et al., firstly documented an enhanced beneath iron flux observed during their field campaign in the Santa Barbara Basin. To further evaluate the impact of such beneath iron flux in the regional biogeochemical cycles, the authors leveraged regional biogeochemical model by adjusting the strength of beneath iron flux. By comparing the model simulations under the different conditions (control vs high flux), the authors found that the beneath iron flux can exert a significant impact in the upper iron concentration and primary productivity. However, in the offshore region, the impact of beneath iron flux on the primary production may be partially counterbalanced by the simultaneously elevated NO3 limitation.

Overall, this manuscript is well written and logically organized, with figures and tables presented in an effective manner. This study provides evaluable insight into how the beneath flux modulates the regional biogeochemistry.

However, I find a severe concern regarding the confidence in the parameterization equation used in the model to simulate high beneath iron flux, which may affect the central results and conclusion in the manuscript. Therefore, I am interested in seeing how authors response before making further recommendation.

**Major Concern:**

The key parameterization assimilated in the biogeochemical model to simulate the high flux scenario relies on the new relationship between the oxygen and beneath iron, which is derived from the synthesis of the new data measured by the authors and historical dataset (Fig. 3 in the main text):

[Figure]

As shown, the newly added data (AT42-19) is quite skewed on the left-hand side, and visually, it gave me a hard time in believing that the inclusion of this new data can cause such a huge difference in the regression equation as presented in present figure, given its limited number of data points and distribution. Are slope and intersect of new equation statistically different with the existing one (i.e. passing one-way ANOVA test?)? Or did authors use the different function to fit the data point? As I mentioned earlier, this new equation is very critical contributing to the key conclusion of your work.

**Other comments:**

Line 89: will the seasonal evolution of oxygen in the beneath be affected by the temporal strength of upper organic carbon export as well?

Figure 1: Did you calibrate the oxygen sensor?  If so, how did you calibrate it? Since the oxygen concentration is quite low on the bottom, the accuracy of sensor is important.

Line 235: it might be helpful to include a map denoting the entire dataset used for fitting the new equation.

Fig. 5: Should use the unit of d-1 for NPP to maintain the consistency with beneath iron flux. Maybe consider including another figure in SI to show the relative percentage change in iron and NPP.

Line 344: The symbol "°" should be included for longitude and latitude numbers.

---

## Author Response (AR1)

**General Response**

We would like to thank both reviewers (RC1 and RC2) for their constructive feedback. Based on their comments, we re-evaluated the model parameterization that links the benthic iron (Fe) flux with bottom oxygen (O2) concentrations, based on the flux compilation by Severmann et al., 2010 and our recent observations. In the re-evaluated parameterization, we use a new model simulation as control, where Fe release follows the exponential fit to all observations, including the new ones. We then use this control to evaluate the effects of local and large-scale Fe inputs. Accordingly, we performed new model simulations to specifically target the impact of high benthic Fe fluxes under low-O2 conditions in the Santa Barbara Basin on surface ocean biogeochemistry and productivity, and compare them to the large-scale Fe supply by atmospheric deposition – the second major external Fe delivery pathway in the region.

Essentially, in the revised manuscript, we compare results of three model simulations. The first (the new baseline Control simulation) is a simulation where we included our new benthic Fe flux measurements from the Santa Barbara Basin to calculate the model benthic Fe fluxes as functions of bottom O2 concentrations. Two other simulations are sensitivity studies based on this Control. In the second simulation (Dust-off) we perturb the system by reducing the dust deposition of Fe to zero everywhere in the model domain. In the third simulation (Hypoxia-off) we cap the benthic Fe fluxes in the Santa Barbara Basin below a maximum value when bottom O2 concentrations are below the hypoxia threshold (60 mmol/m3). A comparison between these simulations and the Control elucidates local and remote effects of external Fe delivery on nutrient cycles and primary production in the California Current. The new results are broadly consistent with those in our prior submission.

Additionally, as pointed out in a prior comment uploaded on the Biogeosciences website, shortly after the submission of our manuscript, we became aware of a recent publication related to our work:

 Wallmann, K., José, Y.S., Hopwood, M.J., Somes, C.J., Dale, A.W., Scholz, F., Achterberg, E.P. and Oschlies, A., 2022. Biogeochemical feedbacks may amplify ongoing and future ocean deoxygenation: a case study from the Peruvian oxygen minimum zone. Biogeochemistry, 159(1), pp.45-67.

We plan to address similarities of this study to ours, including the joint conclusions that ongoing oceanic oxygen loss could drive an increase in benthic iron release, which would enhance local productivity and lead to further oxygen depletion, leading to a positive feedback that could be stabilized by loss of fixed nitrogen under expanded anaerobic conditions.

**Detailed Responses to RC1 (Christopher Somes)**

**RC1 Comment:** Overall I find this to be a useful and interesting study. The paper is nicely structured and well written. The model experiments performed are relevant and well designed. The model experiments certainly show many interesting features demonstrating how benthic iron fluxes are important drivers of NPP in the model, and thus may well be in the real ocean as well. However, I have some reservations about how comprehensively the dynamical numerical model experiments are discussed and support the positive versus dampening feedback mechanisms, from which many of the main conclusions are drawn upon. The most important aspects of the numerical model experiment "High-Flux" are described in only one brief paragraph (Section 3.4). Therefore, I am rather critical of the manuscript in its current form and I think a more robust description of this High-Flux model experiment in particular is necessary before I would endorse it for publication.

**Authors' Response:** We would like to thank Dr. Somes for his positive assessment of the manuscript and his very helpful comments, which help us emphasize an important result of our study. We address all his

comments and concerns in detail, point by point, below. In short, we made the High-flux model simulation our standard (Control) simulation in the revised manuscript, and performed and compared other sensitivity simulations based on and against this High-flux parameterization. We also expanded the description and discussion of our numerical model simulations in both the Results and Discussion sections (Sections 3 and 4).

**RC1 Comment:** line 362-363: "patchwork changes in NPP" Why are the changes in NPP so much patchier in the High-Flux experiment compared to dFe and NO3 changes (Figure 7)?

It is not well explained why the highest hotspots of increased NPP in High-Flux are so far offshore (~36°N, 124°W and 33.5°N,119°W), which does not correspond with the highest increase in dFe around the nearshore zones e.g. Point Conception and the Santa Barbara Channel. For example, is the NPP hotspot immediately west of San Nicolas Island (~33.5°N,119°W) driven by the high dFe from the Santa Barbara basin and horizontal transport or is it derived from benthic dFe release from the deeper Santa Cruz basin and more local vertical transport/mixing?

**Authors' Response:** Note that, as discussed above, the old "High-flux" simulation is now our baseline run, which we compare with the "perturbed" simulations (Hypoxia-off, Dust-off). However, some patchiness in the signals that compare different simulations remain.

We can explain the patchy responses of NPP in the model sensitivity simulations as a result of the changes in iron and nitrogen supply, and the resulting changes in phytoplankton nutrient limitation status and uptake rates. When Fe supply changes in the sensitivity runs, the Fe-limitation status of phytoplankton changes as well. This leads to changes in NPP and NO3- drawdown by phytoplankton, especially near the coast, where the impact of high benthic Fe fluxes is strongest. (It may also alter the proportion of diazotrophs vs. non-diazotrophic phytoplankton, although in the California Current diazotrophs are usually considered to be of minor importance.) Nearshore changes in NO3- concentrations (e.g., in the Santa Barbara Basin), can be transported offshore to impact the nutrient limitation status of phytoplankton and NPP there. For example, a reduction of Fe supply in the Hypoxia-off simulation would increase Fe limitation in and around the Santa Barbara Basin, reducing phytoplankton uptake, and reducing the drawdown of NO3- coastally. Transport of excess NO3- could then fertilize waters offshore. Because the connectivity between coastal waters (including the Santa Barbara Basin) and the offshore region is controlled by mesoscale circulation features such as eddies, some of these impacts can become quite patchy. While we average 6 years of simulation to show model results, this period is likely not long enough to completely smooth-out some of these eddy signals (which would likely require multi-decadal integrations).

We included a more detailed explanation of these issues in Section 3.4 of the revised manuscript.

**RC1 Comment:** Figure 2 shows much higher benthic dFe release at greater depths in the Santa Barbara Basin. Does this trend continue in the deeper basins of the Southern California Bight in the model? I would be curious to see some type of vertical plot (or description at least) of dFe from one of these deeper basins (e.g. Santa Cruz basin near this high NPP hotspot) to see if a significant amount of benthic-derived dFe can avoid being scavenged and make it to the surface ocean in non-coastal settings.

**Authors' Response:** Figure 2 is a direct result of observational data from the Santa Barbara Basin along a single transect sampled during our expedition in fall 2019. The corresponding Fe observations consist of sedimentary fluxes, and do not include water column profiles. However, there is evidence that such high Fe fluxes continue in other anoxic basins of the Southern California Borderline. Severmann et al. (2010) indeed show similarly high fluxes (although smaller than those we measured) for the anoxic or nearly anoxic San Pedro and Santa Monica Basin. Note that suboxia and anoxia in these basins are likely more limited in space and time, and probably less intense, which may lead to somewhat lower Fe release from

the sediment. We also note that the SBB is the shallowest of the 14 basins in the Borderland. Oxygen dynamics, sedimentation and microbial community will also be a factor in determining patterns in benthic flux versus depth.

In terms of Fe profiles, we are not aware of vertical profiles from the region (Santa Cruz or Santa Barbara Basin). Indeed they would be very valuable to constrain the model and the processes at play. The few observations available, e.g., from limited sampling as part of CalCOFI (California Cooperative Oceanic Fisheries Investigations) cruises (King and Barbeau, 2011) mostly focus on the mixed layer, and are too sparse in space and time (they sample selected CalCOFI stations at quarterly intervals) to capture deep water renewal events that are responsible for ventilating the anoxic basins (generally related to strong upwelling events), and presumably allow deep water to be uplifted and eventually transported to the surface by (sub)mesoscale eddies, or sustained upwelling events. Indeed, the model suggests that it is during these ventilation periods that water renewal transports Fe out of the deep basin, which is a prediction that can be tested with dedicated (high frequency) observations, for example before and after renewal events. We plan to add a discussion of previous Fe flux and Fe profile observations in the Southern California Bight to discuss in situ Fe observations more extensively in the revision.

**RC1 Comment:** lines 364-365: "These patterns are opposite in sign to the changes observed in the Low Oxygen Threshold experiments, although more intense, and can be explained by similar dynamics". I suppose this is the reasoning for such a short section 3.4, which in my opinion should be one of the most important in the paper. Figures 6 and 7 do not appear strictly "opposite" to me, although it is difficult to interpret much from Figure 6 with its color bar scale. For example, the main dFe increase in High-Flux (Figure 7) is centered around Point Conception, whereas there appears to be no significant change there in Low Oxygen Threshold-100 (Figure 6).

**Authors' Response**: This is a good observation, with which we agree with. We now clarify these sentences in the results section of the "High-flux" simulation. As mentioned above, the model designs and setups in the revised manuscript will be different compared to the previous version. The impact of high benthic fluxes of Fe from the SBB is explored in the "Hypoxia-off" simulation, while the impact of Fe deposition from dust is explored in the "Dust-off" simulation. Results of these two simulations will still be described in Section 3 of the revised manuscript.

**RC1 Comment:** line 366: "Nearshore, where Fe is more frequently limiting, higher Fe availability releases Fe limitation and drives the higher NPP and more intense NO3 drawdown." But when I look at one of the nearshore regions with the highest increases in dFe in High-Flux, which is centered off Point Conception (Figure 7c), there is very little change if not a slight decrease in NPP there. Thus this claim of widespread Fe limitation in nearshore waters is not very convincing to me. Because if this were true, I would expect a higher correlation between increased nearshore dFe and increased NPP. Thus I think some additional description on this model behavior is required.

**Authors' Response**: The statement that Fe is more frequently limiting nearshore is based on analysis of the model simulations in Deutsch et al., 2020, Progress in Oceanography, see their Fig. 9. However we agree that the statement needs more nuance. In the Deutsch et al., 2020 paper, based on the same model used here, they show that nearshore phytoplankton growth is limited by Fe up to 40% of the time. Further offshore, Fe limitation decreases to about 10% of the time or less. Nitrogen is the limiting nutrient for the remainder of the time. So the reviewer is correct that Fe limitation is not completely widespread nearshore, although it is important during certain times of the year, especially following upwelling. However, based on the model, Fe limitation is indeed more common nearshore than offshore. This is also consistent with observations from King and Barbeau, 2011, who show that N:Fe ratios dramatically decrease moving from the coast to the open ocean, e.g., their Fig. 11. (i.e., N is likely more limiting than Fe offshore). We plan to revise this specific sentence and better contextualize it with references to prior modeling and observational work in revision.

**RC1 Comment:** lines 369-370: "… localized increase in Fe fluxes from the deep SBB has cascading effects on NPP across a much larger region in the CCS."Can the general increase in NPP across the entire Southern California Bight mainly be attributed to the benthic flux from the SBB alone? I would guess increased benthic dFe would occur across the entire Southern California Bight/Borderland region but this is not discussed or shown.

**Authors' Response**: Both in the old "High-flux" simulation, and in the new "Hypoxia-off" simulation, changes in benthic Fe fluxes are applied only to the Santa Barbara Basin (SBB). Thus, the resulting changes in NPP and nutrient cycles relative to the control are solely due to changes in the benthic Fe flux from the SBB. Local perturbations in Fe concentration are then transported upward, reaching the surface (presumably during deep-basin ventilation events, and with participation of sub-mesoscale circulation). The signal is then spread horizontally by the mean and eddy circulation, and thus can affect NPP further downstream. We imagine that the role of the SBB basin is more important than other deep basins, because of the widespread, persistent anoxia and high Fe flux from the sediment from this basin, which is also relatively shallow compared to other low-O2 basins. We plan to add a discussion of these processes, and the relative importance of other Borderland basins relative to the SBB in the revised manuscript. We also plan to add more detail to Section 2.5 (Experimental Design) of the revised manuscript to clarify the set up of the Hypoxia-off experiment.

**RC1 Comment:** Model-Data Comparison (Figure 4). The Control simulation already overestimates offshore dFe concentrations as mentioned in lines 308-309. Since offshore dFe increases even more in High-Flux (Figure 7a), I wonder if surface dFe distribution in High-Flux is improved (or not). The benthic flux measurements indicate that aspect of the model is improved in High-Flux, but could it be that too much dFe is being transported offshore where much of the higher NPP occurs in High-Flux?

**Authors' Response:** In the old manuscript, the Control simulation underestimates dFe concentrations near-shore, and overestimates dFe concentration in the open ocean. In the revised manuscript, we made the "High-flux" our standard simulation and compared the distribution of dissolved Fe in this simulation against available observations. In the new Control, with the higher benthic Fe flux, dFe concentrations increase near-shore, thus improving the model performance. However, a slight increase in the model dFe concentration is also observed in the open-ocean, slightly exacerbating model biases offshore. We plan to expand and revise discussion of these patterns in the Result Section of the new manuscript.

**RC1 Comment:** Spatial Complexities in the Feedback loop. I think the language regarding the positive versus dampening feedback loop could be more specific. For example, Table S1 shows that the High-Flux simulation has exactly (to four significant digits) the same amount of NPP as Control over the full model domain despite that surface dFe is higher on average. Doesn't that suggest that the dampening effect completely compensates for the potential positive feedback over the entire model domain?

I find it pretty remarkable how similar these general results are over the entire model domain compared to my global biogeochemical modeling study where I performed a conceptually similar set model experiments, which tested different benthic and atmospheric Fe fluxes, and also found no significant increase to global NPP despite substantially higher benthic Fe fluxes (Somes et al., 2021). One notable difference is that I increased my scavenging rate constants with source fluxes to prevent my model from overestimating the extent of high dFe concentrations, which also helped my model reproduce the strong offshore dFe gradient. Since it is mentioned that scavenging becomes more rapid when dFe is above the constant 0.6 nM ligand concentration, I wonder if more effective scavenging at high dFe concentrations in High-Flux may contribute to the lack of additional NPP over the entire model domain similar to my global simulations.

On the other hand, it appears that a positive feedback may have developed in the Southern California Bight region, perhaps due to a wider continental shelf/borderland and thus higher dFe fluxes that could more effectively reach the surface there? I think it would be really interesting to have more insights about

how some of these features operate differently throughout the model domain and contribute to the patchy, non-uniform NPP changes in Figure 7c.

**Authors' Response**: Thank you for your thoughtful comments and suggestions. Based on these suggestions, we plan to revise the Discussion Section of the new manuscript, to better evaluate the impact of the positive and damped feedback loop, as well as to link our findings to previous studies. The revised discussion will cover the contents discussed in the following paragraph, and include reference to the global study by Somes et al., 2021:

Essentially, the intense flux of dFe from the sediment suggests the potential for positive biogeochemical feedback in the SBB and more broadly in the CCS (as shown by Figs. 6 – 8). Under a positive feedback scenario (e.g., reflecting the schematic of Fig. 9a), anoxic and nearly anoxic bottom water conditions facilitate Fe(II) diffusion from the sediment into the bottom water. In the SBB, this Fe eventually reaches the surface via upwelling and mixing processes, which are likely enhanced in the presence of complex bathymetry and islands, as observed in the Southern California Bight region in our model simulations. This additional dFe input fertilizes coastal waters and increases primary production. Newly formed organic matter eventually sinks towards the seafloor as a rain of organic particles, supporting low-oxygen concentrations in the bottom water, and fueling anaerobic respiration, including Fe reduction, in the sediment. This chain of processes thus represents a positive feedback loop that maintains high Fe(II) release from the sediment, as long as the bottom water remains hypoxic or anoxic. Our simulations also indicate the presence of complex biogeochemical responses between Fe, NO3 and NPP, which can dampen the effects of these feedbacks. Specifically, the positive feedback loop in our model simulation is dampened by an increased NO3- limitation under higher Fe supply (following the schematic of Fig. 9b), which would in turn limit the increase in NPP. Transport of N-depleted coastal waters can further reduce NPP offshore, counteracting the positive feedback. In addition, the positive feedback could be also damped by Fe scavenging in our model, as the Fe scavenging rate is set to rapidly increase when the dissolved Fe concentration is above a constant threshold 0.6 nM (as discussed in Section 2). This damping effect by scavenging on the impact of high benthic Fe fluxes has been documented at the global scale by the modeling study by Somes et al., 2021.

**General Response**

We would like to thank both reviewers (RC1 and RC2) for their constructive feedback. Based on their comments, we re-evaluated the model parameterization that links the benthic iron (Fe) flux with bottom oxygen (O2) concentrations, based on the flux compilation by Severmann et al., 2010 and our recent observations. In the re-evaluated parameterization, we use a new model simulation as control, where Fe release follows the exponential fit to all observations, including the new ones. We then use this control to evaluate the effects of local and large-scale Fe inputs. Accordingly, we performed new model simulations to specifically target the impact of high benthic Fe fluxes under low-O2 conditions in the Santa Barbara Basin on surface ocean biogeochemistry and productivity, and compare them to the large-scale Fe supply by atmospheric deposition – the second major external Fe delivery pathway in the region.

Essentially, in the revised manuscript, we compare results of three model simulations. The first (the new baseline Control simulation) is a simulation where we included our new benthic Fe flux measurements from the Santa Barbara Basin to calculate the model benthic Fe fluxes as functions of bottom O2 concentrations. Two other simulations are sensitivity studies based on this Control. In the second simulation (Dust-off) we perturb the system by reducing the dust deposition of Fe to zero everywhere in the model domain. In the third simulation (Hypoxia-off) we cap the benthic Fe fluxes in the Santa Barbara Basin below a maximum value when bottom O2 concentrations are below the hypoxia threshold (60 mmol/m3). A comparison between these simulations and the Control elucidates local and remote effects of external Fe delivery on nutrient cycles and primary production in the California Current. The new results are broadly consistent with those in our prior submission.

Additionally, as pointed out in a prior comment uploaded on the Biogeosciences website, shortly after the submission of our manuscript, we became aware of a recent publication related to our work:

Wallmann, K., José, Y.S., Hopwood, M.J., Somes, C.J., Dale, A.W., Scholz, F., Achterberg, E.P. and Oschlies, A., 2022. Biogeochemical feedbacks may amplify ongoing and future ocean deoxygenation: a case study from the Peruvian oxygen minimum zone. Biogeochemistry, 159(1), pp.45-67.

We plan to address similarities of this study to ours, including the joint conclusions that ongoing oceanic oxygen loss could drive an increase in benthic iron release, which would enhance local productivity and lead to further oxygen depletion, leading to a positive feedback that could be stabilized by loss of fixed nitrogen under expanded anaerobic conditions.

**Detailed Responses to RC2 (Anonymous Reviewer)**

**RC2 Comment:** I find a severe concern regarding the confidence in the parameterization equation used. in the model to simulate high beneath iron flux, which may affect the central results and conclusion in the manuscript. Therefore, I am interested in seeing how authors respond before making further recommendations.

The key parameterization assimilated in the biogeochemical model to simulate the high flux scenario relies on the new relationship between the oxygen and beneath iron, which is derived from the synthesis of the new data measured by the authors and historical dataset (Fig. 3 in the main text):

As shown, the newly added data (AT42-19) is quite skewed on the left-hand side, and visually, it gave me a hard time in believing that the inclusion of this new data can cause such a huge difference in the

regression equation as presented in present figure, given its limited number of data points and distribution. Are slope and intercept of the new equation statistically different with the existing one (i.e. passing one-way ANOVA test?)? Or did authors use a different function to fit the data point? As I mentioned earlier, this new equation is very critical contributing to the key conclusion of your work.

**Authors' Response**: We would like to thank Reviewer 2 for this comment, which helped us to substantially revise and clarify our study. In the old (submitted) manuscript, we used the parameterization for benthic Fe fluxes as a function of bottom O2 concentrations derived in a previously published version of ROMS-BEC by Deutsch et al., 2020, based on Severman's 2010 flux measurements. The model has been evaluated against observations and has been used in ocean biogeochemical studies for the U.S. West Coast. However, based on the Reviewers' concerns, we decided to change our approach, and re-calculated the best exponential fit between bottom O2 and benthic Fe fluxes from Severmann et al., (2010)'s data, with and without inclusion of our new data from the AT42-19 cruise (see included Figure) .

[Figure]

Comparing this re-calculated best fit line for Severmann's data only (yellow line) with the best fit line for the combine dataset with both Severmann's and AT42-19 cruise's data (purple line), we see that with the inclusion of new data, the slope of the best fit line gets steeper, which means that we have higher benthic Fe fluxes at low values of bottom oxygen concentrations (<120 µmol/kg) and lower benthic Fe fluxes at high values (> 140 µmol/kg). Note that oxygen concentrations in the deep basin of the Santa Barbara are always below 120 µmol/kg, which leads to a higher benthic Fe flux in this region and, in the model simulations, to higher surface dissolved Fe concentrations.

As discussed earlier in the response document, to avoid any confusion, we revised all model simulations used in this study to apply the exponential fit to the combined Fe flux data (Severmann et al., 2020, plus AT42-19) as our default Control simulations (purple line in the attached figure), and use that as starting point for the perturbations in the Hypoxia-off and Dust-off simulations.

**RC2 Comment:** Line 89: will the seasonal evolution of oxygen beneath be affected by the temporal strength of upper organic carbon export as well?

**Authors' Response:** Yes, we would expect oxygen evolution to be affected by seasonal changes in organic carbon export – insofar as that affects bottom O2 via remineralization. Export in turn is driven by surface processes such as nutrient availability, NPP, and phytoplankton size. Note that the model does a reasonable job capturing these processes (e.g. as discussed in Deutsch et al., 2020)

**RC2 Comment:** Figure 1: Did you calibrate the oxygen sensor? If so, how did you calibrate it? Since the oxygen concentration is quite low on the bottom, the accuracy of the sensor is important.

**Authors' Response:** The oxygen sensor (Aandara Optode w/ fast foil) mounted to the AUV Sentry was calibrated by the manufacturer prior to the expedition. Note that we do have several lines of evidences that confirm anoxia in the deeper parts of the SBB during our expedition, which includes the AUV Sentry optode, the ROV Jason optode, Winkler titration from CTD/Rosette casts, benthic chamber optodes, and benthic in-situ microprofiling. Details are provided in a separate study currently under review at Biogeosciences:

Yousavich, D. J., Robinson, D., Peng, X., Krause, S. J. E., Wenzhoefer, F., Janßen, F., Liu, N., Tarn, J., Kinnaman, F., Valentine, D. L., and Treude, T.: Marine anoxia initiates giant sulfur-bacteria mat proliferation and associated changes in benthic nitrogen, sulfur, and iron cycling in the Santa Barbara Basin, California Borderland, EGUsphere [preprint], https://doi.org/10.5194/egusphere-2023-1198, 2023.

**RC2 Comment:** Line 235: it might be helpful to include a map denoting the entire dataset used for fitting the new equation.

**Authors' Response:** Thank you for your suggestion. We will add such a figure to the revised manuscript.

**RC2 Comment:** Fig. 5: Should use the unit of d-1 for NPP to maintain the consistency of the beneath iron flux. Maybe consider including another figure in SI to show the relative percentage change in iron and NPP.

**Authors' Response:** Thank you for your suggestion. We will change the unit for NPP in the revised manuscript and add an additional figure in SI to show the relative percentage change.

**RC2 Comment:** Line 344: The symbol "°" should be included for longitude and latitude numbers.

**Authors' Response:** Thank you for pointing out our omission. We will add the degree symbol at the end of the Lat/Long numbers.